# Reasons for Surgery Cancellation in a General Hospital: A 10-year Study

**DOI:** 10.3390/ijerph16010007

**Published:** 2018-12-20

**Authors:** Hyun Sun Cho, Ye Seol Lee, Sang Gyu Lee, Ji Man Kim, Tae Hyun Kim

**Affiliations:** 1PI (Performance Improvement) Team, Dongguk University Ilsan Hospital, Gyeongi-Do 10326, Korea; cpr1309@gmail.com; 2Department of Hospital Administration, Graduate School of Public Health, Yonsei University, Seoul 03722, Korea; leevan@yuhs.ac; 3Department of Public Health, Graduate School, Yonsei University, Seoul 03722, Korea; yeseol.lee127@gmail.com; 4Institute of Health Services Research, Yonsei University, Seoul 03722, Korea; 5Health Insurance Review and Assessment Service, Wonju 26465, Korea; mann25@gmail.com

**Keywords:** surgery cancellation, elective surgery, emergency surgery

## Abstract

*Background*: This study researched related causes that make scheduled surgeries canceled not to be conducted and based on the research it is to derive issues in order to reduce surgery cancellation. *Methods*: We analyzed the association of surgery cancellation with patient characteristics, surgical characteristics and surgery schedule related characteristics, using electronic medical record (EMR) data on surgeries conducted at a university hospital in Korea over 10 years. Additionally, we examined the reasons for surgery cancellation based on patient and hospital characteristics. We used chi-square tests to analyze the distribution of various characteristics according to reasons for surgery cancellation. Multivariate logistic regression analyses were conducted to evaluate the factors associated with surgery cancellation. *Results*: Among 60,333 cases, surgery cancellation rate was 8.0%. The results of the logistic regression indicated a high probability of surgery cancellation when the patient was too old (odds ratio [OR]: 1.35, 95% confidence interval [CI]: 1.14–1.59), when it was a neurosurgery case (OR: 1.39, 95% CI: 1.21–1.59), when regional anesthesia was used (OR: 1.15, 95% CI: 1.07–1.24) or when it was a planned surgery (OR: 2.45, 95% CI: 2.21–2.73). The surgery cancellation rate was lower when the patient was female (OR: 0.87, 95% CI: 0.82–0.93) or when the surgery was related to Obstetrics & Gynecology (OR: 0.53, 95% CI: 0.46–0.60) or Ophthalmology (OR: 0.66, 95% CI: 0.56–0.79). Among the canceled 4834 cases, the surgery cancellation rate for the reasons of patients was 93.2% and the surgery cancellation rate for the reasons of a hospital was 6.8%. *Conclusions*: This study found that there are related various causes to cancel operations, including patient characteristics, surgery related characteristics and surgery schedule related characteristics and it means that it would be possible for some reasons to be prevented. Every medical institution should consider the operation cancellation as an important issue and systematic monitoring should be needed.

## 1. Introduction

Operating rooms form a key area of service within hospitals for patient care, with the highest initial investment cost per unit area and an extensive amount of time required for training of the workforce. Therefore, efficient management of operating rooms has direct effects on the surgery of hospitals [1]. Simply increasing the number of operating rooms to accommodate the rising number of surgeries is not a cost-efficient method and, consequently, there have been various suggestions for more efficient management of operating rooms. In this situation, there is a need to determine an effective method to improve the operating room turnover time in order to maximize the efficiency of operating room management without additional facility investment [2].

Several services provided by hospitals need prior appointment or reservation and efficient management of these appointments or reservations allows smooth coordination of the workload between the diagnostic and supporting departments. Surgery is no exception, with different departments providing diverse surgeries for numerous patients. However, surgery cancellations often occur, which is a major problem that most hospitals are yet to resolve satisfactorily. One of the major issues arising due to surgeries being cancelled on the day of surgery is that the workforce, surgical tools and space for the planned surgery are unable to be utilized, which results in wasted hospital resources. Furthermore, cancelled surgeries lead to an accumulation of surgeries and reduce the number of patients who could otherwise undergo scheduled surgeries and then be discharged on time, thereby limiting the number of new patient admissions [3]. Finally, cancelled surgeries have psychological and economic effects on both the patients and their families. Overall, surgery cancellations have negative effects on both hospitals and patients. Therefore, efficient management of operating rooms to reduce the number of surgery cancellations is crucial for both the efficient surgery of hospitals and for a positive experience for patients [4].

As shown in previous studies, surgery cancellation rates between different hospitals vary between 5% and 20% [3,4,5,6,7,8]. These results indicate that, while some hospitals manage a low rate of surgery cancellations, there are other hospitals exhibiting a high rate of surgery cancellations. Hospitals with high rates of surgery cancellations need to pay more attention and place greater efforts on minimizing the number of surgery cancellations. Therefore, an approach that analyzes the factors and reasons involved for surgery cancellations is needed to reduce the rate of surgery cancellations at a hospital level. The rate of surgery cancellations and the efficient use of hospital facilities and resources are closely related. Indices to assess the efficiency of operating room management include overtime costs, delayed initiations of surgery, rates of cancellation, delayed admissions to the recovery room after surgery, delays to the next surgery after a previous surgery and errors in the estimated duration of surgery [9]. Among these indices, cancellations on the day of surgery remain an unresolved issue in medical institutions worldwide [10]. Reducing the rate of cancellations on the day of surgery can dramatically increase hospital revenue and decrease resource waste; therefore, hospitals need to provide organization-level efforts and resource management to reduce surgery cancellations [10]. There is a need for a thorough analysis of the rate of cancellations and of the factors leading to surgery cancellations for both general and emergency surgeries, as well as for an analysis of the reasons for cancellations in respect of both hospitals and patients.

This study aimed to investigate the factors related to surgery cancellation and the reasons for the same. Additionally, the relationship of surgery cancellation and its reasons with hospital and patient factors were examined separately.

## 2. Methods

### 2.1. Data Collection

The present study utilized data from the electronic medical record (EMR) of a university hospital located in Gyeonggi Province, Korea. The study site is a general hospital with about 1000 beds and it is a large-scale medical center that conducts medical research as well as patient treatment. Surgery-related EMR data on 85,846 cases between 1 January 2007 and 31 December 2016 were extracted. The following surgeries were excluded from the analysis: surgeries planned on Saturdays and holidays; local anesthesia-based surgeries (and procedures); Electroconvulsive therapy (ECT) from the department of psychiatry; procedures performed at the department of anesthesiology; and surgeries performed at the department of dentistry. The final dataset included 60,330 expected surgery cases of 28,851 males and 31,479 females (Figure 1).

Data on patients’ Korean standard classification of diseases-7 (KCD-7) codes used to classify pre-operative diagnostic variables and the status of two chronic diseases (i.e., hypertension, diabetes) that could not be extracted from the electronic records system, were obtained from the department of medical record. 

### 2.2. Variables

Surgery cancellation was defined as a case where a surgeon entered a planned surgery date into the surgery schedule-assistant program, the department of anesthesiology approved the surgery schedule and medical resources (operation room and workforce) were ready but the surgery was cancelled on the day of surgery by either the hospital or the patient. 

The general patient characteristics were independent variables and included sex, age, admission status and chronic diseases. Patients were divided into 5 groups based on age (≤19 years, 20–39 years, 40–59 years, 60–79 years and ≥80 years). For admission status, the patients were divided into a day surgery group and an inpatient surgery group admitted for 2 days or longer. The presence of chronic diseases, hypertension and diabetes, were assessed in the patients. 

Since blood sugar and blood pressure levels may affect not only the anesthetic management before the surgery but also the medical outcomes after the surgery, patients’ information about hypertension and diabetes need to be considered [11]. We divided patients into the following 4 groups: patients with neither disease and those with diabetes, hypertension or both diseases.

For surgery-related factors, the patients were divided based on the department associated with the surgery, anesthesia type and emergency status. The patients were divided into the following 9 groups based on the following departments associated with the surgery: Orthopedic Surgery (OS); General Surgery (GS); Obstetrics and Gynecology (OB & GY); Neurosurgery (NS); Thoracic and Cardiovascular Surgery (CS); Plastic Surgery (PS); Ear, Nose & Throat (ENT); Ophthalmology (EY); and Urology (UR). For the division based on anesthesia type, the patients were divided into either a general anesthesia group or a regional anesthesia group. In this study, spinal anesthesia, epidural anesthesia and nerve block anesthesia were considered as regional anesthesia. Surgeries can largely be divided into planned and emergency surgeries. Planned surgery was defined as a planned surgery with appropriate pre-operative examinations performed by a surgeon, while emergency surgery involved an urgent surgery performed, as determined by the surgeon, without adequate pre-operative examinations in cases where the surgeon believed that urgent surgery would provide the best chance of survival for the patient. There were 18 pre-operative diagnoses using Korean standard classification of diseases-7 (KCD-7).

For surgery schedule characteristics, the year, season and date of the surgery were included. The patients in this study underwent surgery between 1 January 2007 and 31 December 2016 and were divided into 10 groups based on the year of surgery. The seasons were defined as follows and the following 4 groups were formed: March to May as spring; June to August as summer; September to November as autumn; and December to February as winter. For the surgery date, there were 5 different groups, with one group per week day, excluding Saturday and Sunday. 

The reasons for surgery cancellation were classified into patient and hospital factors. Patient factors included receiving additional treatment and screening before surgery for sudden changes in the clinical condition, refusing surgery, delay due to the patient, incomplete surgical workup because of taking medication that the patient was not supposed to take before the surgery or not fasting before the surgery, change in medical status or treatment plan and others. Hospital factors included surgeon unavailability, operation in another place, scheduling error, lack of beds, unavailability of equipment, incomplete medical evaluation and others. 

### 2.3. Statistical Analysis

General, surgical and operation schedule characteristics are shown using descriptive statistics including frequency and percentage values. Surgery cancellation was considered as a binary variable and a Chi-square test was performed to assess the association between surgery cancellation and each of the variables. Cancelled surgeries were divided into two groups based on who cancelled the surgery, whether the hospital or the patient and frequency and percentage values were used for the analysis.

A multivariable logistic regression model was used to assess the factors associated with surgery cancellations. Pre-operative diagnosis was excluded from the analysis due to multicollinearity. Additionally, we identified important determinants through the stepwise selection function in the model options. Subgroup analysis was performed according to the urgency of the surgery. Data analysis was performed using SAS software (ver. 9.4; SAS Institute, Cary, NC, USA). Our methods were guided by Strengthening the Reporting of Cohort Studies in Surgery (STROCSS) criteria [12]. The Institutional Review Board (IRB) of Dongguk University Ilsan Hospital approved this study (IRB approval; DUIH 2017-06-003-002-HE002).

## 3. Results

### 3.1. Comparison of Characteristics Between Performed and Cancelled Surgeries

From the entire patient of 60,330 surgeries, there were 55,496 performed surgeries and 4834 cancelled surgeries, with an annual average cancellation rate of approximately 8.0% over 10 years. According to general patient characteristics, 9.0% of cancelled surgeries involved male patients and 7.1% involved female patients. Regarding age groups, the ≥80-year-old group exhibited an 11.6% cancellation rate. In terms of inpatients (≥2 days admission), 8.2% had cancelled surgeries and the cancellation rate was 10.2% in patients with both hypertension and diabetes.

For surgical characteristics, the department of neurosurgery had the highest cancellation rate of 10.5%, followed by urology (10.4%) and orthopedic surgery (9.7%). Based on the type of anesthesia, the cancellation rate was higher for the regional anesthesia group (9.5%) than for the general anesthesia group (7.5%). The cancellation rate was higher for planned surgery (8.8%) than for emergency surgery (4.1%) and, based on pre-operative diagnosis, the following groups had the highest cancellation rates in order: ‘Symptoms, signs and abnormal clinical and laboratory findings, NEC (R00-R99)’, 23.2%; ‘Certain infectious and parasitic diseases (A00-B99)’, 15.6%; and ‘Endocrine, nutritional and metabolic diseases (E00-E90)’, 15.4%. Based on surgery schedule characteristics, the cancellation rate was higher on Mondays (9.1%) than on the other days (Table 1).

### 3.2. Analysis of the Reasons for Cancellation

The reasons for cancellation are grouped, based on who cancelled the surgery, in Table 2. Of 4834 cancelled surgeries 4505 (93.2%) were cancelled by the patients and 329 (6.8%) were cancelled by the hospital. The following were the main reasons for cancellation by patients: 1566 surgeries were cancelled (32.4%) because of patient refused surgery; and 950 (19.7%) because of delay due to personal reasons. The main reasons for cancellations by the hospital included incomplete medical evaluation (161 cases, 3.3%) and cancellation due surgeon not available (65 cases, 1.3%). Considering that surgeries cancelled by the hospital are preventable and that 329 out of 60,330 cases were cancelled by the hospital, there is scope to prevent approximately 33 surgery cancellations annually.

### 3.3. Multivariate Analysis on Surgery Cancellation Factors

#### 3.3.1. All Surgeries

Multivariable logistic regression results of the factors associated with surgery cancellations are outlined in Table 3. Based on sex, the odds ratio was significantly lower in females (OR = 0.87, 95% CI = 0.82–0.93) than in males, indicating that female patients had a lower likelihood of surgery cancellation than male patients. Based on age groups, the odds ratios for the group aged between 20 and 39 years (OR = 0.76, 95% CI = 0.68–0.85) and the group aged between 40 and 59 years (OR = 0.89, 95% CI = 0.80–0.98) were significantly lower than for the group aged ≤ 19 years, while the odds ratio for the group aged ≥ 80 years (OR = 1.35, 95% CI = 1.14–1.59) was significantly higher than for the group aged ≤ 19 years. Based on admission status, the odds ratio for inpatient surgeries(OR = 2.71, 95% CI = 2.03–3.61) was significantly higher than for outpatient or day surgeries, indicating that admitted patients had a higher risk of surgery cancellation. For chronic diseases, the odds ratio for patients with hypertension (OR = 0.90, 95% CI = 0.82–0.99) was significantly lower than for patients with neither disease.

Based on the department that performed surgery, the odds ratios for surgery cancellations for obstetrics and gynecology (OR = 0.53, 95% CI = 0.46–0.60), ear, nose & throat (OR = 0.75, 95% CI = 0.68–0.84) and ophthalmology (OR = 0.66, 95% CI = 0.56–0.79) were significantly higher than for orthopedic surgery but the odds ratio for neurosurgery (OR = 1.39, 95% CI = 1.21–1.59) was significantly lower than for orthopedic surgery. For anesthesia type, the odds ratio for regional anesthesia (OR = 1.15, 95% CI = 1.07–1.24) was significantly higher than for general anesthesia, indicating a higher risk of surgery cancellation for patients undergoing regional anesthesia. Finally, planned surgery had a much higher odds ratio (OR = 2.45, 95% CI = 2.21–2.73) than emergency surgery and the results were statistically significant, indicating that patients planning to undergo planned surgery had a higher risk of surgery cancellation. 

From surgery schedule characteristics, both years and seasons were not associated with a risk of surgery cancellation. However, patients with planned surgery on Mondays had a significantly higher odds ratio than those with planned surgery on Tuesdays (OR = 0.79, 95% CI = 0.72–0.87), Wednesdays (OR = 0.87, 95% CI = 0.79–0.96), Thursdays (OR = 0.86, 95% CI = 0.79–0.95) or Fridays (OR = 0.86, 95% CI = 0.78–0.94), indicating that the risk of surgery cancellation on other weekdays is relatively lower than that on Mondays. 

#### 3.3.2. Subgroup Analysis by the Type of Surgery

Multivariable logistic regression was performed on subgroups divided based on the urgency of the surgery and the results for the planned surgery subgroup are outlined in Table 4. Females had a significantly lower odds ratio (OR = 0.88, 95% CI = 0.82–0.94) than males, suggesting that females had a lower risk of surgery cancellation. For age groups, the odds ratios for the group aged between 20 and 39 years (OR = 0.76, 95% CI = 0.68–0.85) and for the group aged between 40 and 59 years (OR = 0.86, 95% CI = 0.77–0.96) were lower compared to the group aged ≤ 19 years. The odds ratio for the patients having inpatient surgery was significantly higher (OR = 3.24, 95% CI = 2.37–4.44) than for those having outpatient surgery or day surgery, indicating that admitted patients (≥2 days of admission) had a higher risk of surgery cancellation.

Based on the department that performed the surgery, the odds ratios for obstetrics and gynecology (OR = 0.53, 95% CI = 0.46–0.61), ear, nose & throat (OR = 0.75, 95% CI = 0.67–0.84) and ophthalmology (OR = 0.68, 95% CI = 0.57–0.82) were significantly higher than for orthopedic surgery but the odds ratio for neurosurgery (OR=1.63, 95% CI = 1.41–1.89) was significantly lower than for orthopedic surgery. For anesthesia type, the odds ratio for regional anesthesia (OR = 1.14, 95% CI = 1.05–1.23) was significantly higher than for general anesthesia, indicating a higher risk of surgery cancellation for patients undergoing regional anesthesia.

From surgery schedule characteristics, the patients with planned surgery on Mondays had a significantly higher odds ratio than those with planned surgery on Tuesdays (OR = 0.80, 95% CI = 0.73–0.88), Wednesdays (OR = 0.90, 95% CI = 0.82–0.99), Thursdays (OR = 0.89, 95% CI = 0.81–0.98) or Fridays (OR = 0.88, 95% CI = 0.80–0.97), indicating that the risk of surgery cancellation on other weekdays is relatively lower compared to Mondays.

The results for emergency surgery subgroup are outlined in Table 5. Based on the age groups, the odds ratios for the group aged between 60 and 79 years (OR = 1.69, 95% CI = 1.17–2.45) and the group aged ≥ 80 years (OR = 2.93, 95% CI = 1.83–4.69) were higher than for the group aged ≤ 19 years. The odds ratio for the patients having inpatient surgery was lower (OR = 0.42, 95% CI = 0.21–0.86) than for those having outpatient surgery or day surgery.

Based on the department that performed surgery, the odds ratios for general surgery (OR = 0.37, 95% CI = 0.27–0.51), obstetrics and gynecology (OR = 0.44, 95% CI = 0.27–0.70) and neurosurgery (OR = 0.53, 95% CI = 0.37–0.78) were significantly lower than for orthopedic surgery.

From surgery schedule characteristics, the patients with surgeries on Mondays had a significantly higher odds ratio than those with surgeries on Wednesdays (OR = 0.62, 95% CI = 0.46–0.84), Thursdays (OR = 0.66, 95% CI = 0.49–0.90) or Fridays (OR = 0.65, 95% CI = 0.48–0.88), indicating that the risk of surgery cancellation on the above weekdays was relatively lower compared to Mondays.

## 4. Discussion

The primary purpose of managing surgery cancellation rates is to reduce cancellation rates and improve the efficiency of operating room management, through analyzing factors associated with surgery cancellations and taking preventive measures. This study provides fundamental data for the management of surgery cancellations, utilizing a relatively large dataset of 60,330 surgery cases between 1 January 2007 and 31 December 2016, extracted from the electronic medical records of a university hospital located in Gyeonggi province.

Previous studies assessing surgery cancellation rates and the factors associated with surgery cancellations in patients with planned surgeries performed simple analyses of the relationship between the cancellation rate and other variables, including sex, age, department performing the surgery, day and month [3,7,13,14]. However, in this study, a comprehensive analysis of the factors associated with surgery cancellations was performed through grouping variables into general, surgical and operation schedule characteristics and including additional variables such as admission status, chronic diseases, anesthesia type, emergency status, pre-operative diagnosis and year and season when the surgery was planned. Moreover, the majorities of previous studies only included planned surgeries and excluded emergency surgeries but in this study both emergency and planned surgeries were included. In addition, patients were then put into different subgroups based on surgery type (general and emergency) and multivariable regression analysis was separately performed on the two subgroups to assess the factors in respect of surgery cancellation for each subgroup. Furthermore, the reasons for cancellation were divided into cases cancelled by patients and by the hospital, respectively, to provide fundamental data for implementing preventive measures to address the reasons that lead to surgery cancellations. The evidential and analytical methods used in this study have been more comprehensive than those of other studies assessing surgery cancellation rates and the related reasons.

From this study, the analysis results on the characteristics and related factors in respect of operation cancellations were as following: of 60,330 patients with planned surgery, 55,496 patients underwent the planned surgery while 4834 patients canceled the surgery, with a cancellation rate of 8.0%. The rate of cancellation for planned surgery was 8.8% and 4.1% for emergency surgery.

On the other hand, the cancellation rate identified by Bae [15] was 10.7% and other studies from well-established medical institutions in the U.S. and the U.K. have identified rates of 5.19% and 9.3%, respectively [10,16]. Another study performed in Brazil showed the cancellation rate of 16.1% [17]. Moreover, from studies that only assessed planned surgery and excluded emergency surgery, diverse cancellation rates of 3.49%, 5.13%, 6.9% and 16.2% were obtained [4,13,14]. A study by Kaddoum et al. reported that the cancellation rate of planned surgery was 4.4% [18]. The cancellation rates in previous studies that included emergency surgery also differed widely, with rates between 5.19% and 16.1% identified. Due to the differences in the study periods and in the patients involved, it is difficult to make a direct comparison with the results of previous studies. There has been no previous study with an identical experimental design and thus it is difficult to make a direct comparison of these results. However, the cancellation rate identified in this study approximated the average cancellation rate identified in studies that included emergency surgery in their analysis. 

Conventional studies have usually analyzed the surgery cancellation rate using simple descriptive statistics and univariate analysis, limiting the number of studies that have assessed factors associated with surgery cancellation. The results from limited studies that have assessed delayed surgeries or cancellations of planned outpatient surgery were also considered and their results compared with the results of this study.

In the following sections, comparisons of the factors associated with surgery cancellation in this study and those factors identified in previous studies are summarized. First, based on sex, the risk of surgery cancellation was higher in males than in females for all surgeries and planned surgeries but the odds ratio difference was insignificant between sexes for emergency surgeries. On the other hand, a study by Kim (2002) [2] that assessed delayed operating room occupancy reported a higher risk of delayed operating room occupancy for female patients compared to male patients and a study by Sung et al. (2010) [19] reported that there was no significant difference in the surgery cancellation rate based on sex. Another study by Kwon et al. (2015) [20] that assessed cancellation of planned outpatient surgery supported the finding that sex and cancellation rates were not associated. The odds ratio for inpatient (≥2 days admission) surgery was significantly higher than for that of outpatient or day surgery for all surgeries and planned surgeries but significantly lower for emergency surgeries. In contrast, the study by Sung et al. (2010) [19] reported a higher odds ratio of surgery cancellations for outpatient surgeries than for inpatient surgeries. The difference in the cancellation rate in this study compared to those reported in previous studies was likely because we excluded surgeries with local anesthesia and hence excluded the majority of outpatient surgeries. From the study by Kim (2002) [2] that assessed delayed operating room occupancy, it was reported that there was no association between admission status and delayed occupancy of operating rooms.

Second, based on surgical characteristics, the surgery cancellation rates were lower for general surgery, obstetrics and gynecology, ear, nose & throat and ophthalmology than for orthopedic surgery. However, the cancellation rate was higher for neurosurgery than for orthopedics. A study by Cihoda and Alves (2015) [17] also reported a significantly higher surgery cancellation rate in orthopedic surgery and general surgery. Sung et al. (2010) [19] reported that the cancellation rate was lower for neurosurgery, plastic surgery and general surgery than for obstetrics and gynecology, while the odds ratio for ophthalmology surgery was high. The cancellation rates for emergency surgery were higher in general surgery, obstetrics and gynecology and neurosurgery than for orthopedic surgery. This is likely due to a higher incidence of emergency surgeries in respect of general surgery, obstetrics and gynecology and neurosurgery and that emergency surgeries are less likely to be cancelled. For all surgeries and planned surgeries, the odds ratio for operation cancellations was significantly higher for regional anesthesia than for general anesthesia but there was no difference for emergency surgeries. These results are in agreement with the previous study by Kim (2002) [2] that reported higher odds ratio for delayed occupancy of operating rooms for regional anesthesia patients than for general anesthesia patients. For general anesthesia, basic examinations of ECG, chest radiographs and blood tests are performed for all surgeries prior to surgery, while examinations are often omitted in patients receiving regional anesthesia surgeries. Therefore, surgery cancellations are more likely to occur for regional anesthesia surgeries. The odds ratio for cancellation was significantly higher for planned surgeries than for emergency surgeries. The study by Kim (2002) [2] supported these findings although the results of that study were not statistically significant. Emergency surgeries are less likely to be cancelled as patients receiving emergency surgeries are often in a critical condition and require immediate treatment.

Third, the cancellation rates based on surgery schedule characteristics were different, with the highest cancellation rates for surgeries planned on Mondays. These results are in agreement with the results from the study by Kwon et al. (2015) [20] that assessed cancellation rates of outpatient surgery, where the authors reported a higher likelihood of cancellations on Mondays. On the other hand, the study by Kim (2002) [2] demonstrated a higher risk of delayed operating room usage on Thursdays. In general, patients with planned surgery are admitted a day before surgery to receive pre-operative treatment and receive surgery on the following day. If the surgery is planned on Monday, pre-operative treatments need to be performed on Sunday and the operating surgeon is often not in the hospital on Sundays. Furthermore, cooperative diagnosis systems linked with other departments are only available for emergency cases on Sundays. Therefore, it is more difficult to assess the condition of patients admitted on Sundays than for those admitted on weekdays, thereby increasing the risk of surgery cancellation on Mondays.

From the analysis of the reasons for cancellations, 93.2% of the operations were cancelled by the patients and only 6.8% of the cases were cancelled by the hospital. The most frequent reason for cancellation was ‘the need for additional pre-operative examination and treatment’ involving 32.4%, followed by ‘refusal to receive surgery or not hospitalized’ involving 29.9% and ‘delayed operation due to personal reasons’ involving 19.7%. The reasons for cancellation were similar for planned surgeries. These results are in agreement with numerous previous studies that report ‘underlying diseases or issues in diagnosis’ and ‘not admitted’ as the main reasons for cancellation of planned surgeries [4,7,13].

However, there are some limitations to this study. First, since the study cohort comprised patients with planned surgeries from a single university hospital, it is difficult to generalize the findings to all university hospitals. Therefore, a larger-scale future study on patient cohort data collected from other hospitals is required. Second, this study is a cross-sectional study and cannot confirm causal relationships between surgery cancellations and other factors, such as the department that performed the surgery, anesthesia type and emergency status. Third, medical records on chronic diseases were unavailable for cardiovascular or cerebrovascular diseases, chronic respiratory diseases and cancer. It was only possible to obtain information on hypertension and diabetes from the medical records library and therefore the analysis of chronic diseases as an independent variable was of limited value. Fourth, emergency surgeries are performed in cases where a lack of immediate surgical intervention often leads to poorer prognosis for patients but emergency surgeries included in this study were based on the emergency need not only of the patient but also on the availability of the operating surgeon. An emergency for the operating surgeon was defined as a case where the patient was able to receive surgical treatment a day or two later but due to the operating surgeon’s schedule (i.e., planned outpatient clinic or travelling to conferences) the surgery had to be performed on the same day. Therefore, in future, the emergency status based on the patient’s condition must be clearly delineated and the emergency situation of the operating surgeon needs be separately classified to reflect the availability of the surgeon in cases where the patient is not otherwise in an urgent need for a surgery. 

## 5. Conclusions

In this study, factors associated with surgery cancellations and the reasons for these cancellations were identified and analyzed. To reduce cancellation rates of planned surgeries, continuous monitoring of cancellation rates and quality control systems to accurately analyze the reasons for cancellations and address cancellation rates are necessary. Complete and accurate assessment of a patient’s condition may reduce the surgery cancellation rates due to medical issues, while the establishment of personal relationships with patients and the introduction of a pre-operative Reminder Call system may reduce cancellation rates due to non-medical issues. Additional in-depth studies are required to establish more detailed operating room management systems that control for factors associated with surgery cancellations, based on the findings of this study.

## Figures and Tables

**Figure 1 ijerph-16-00007-f001:**
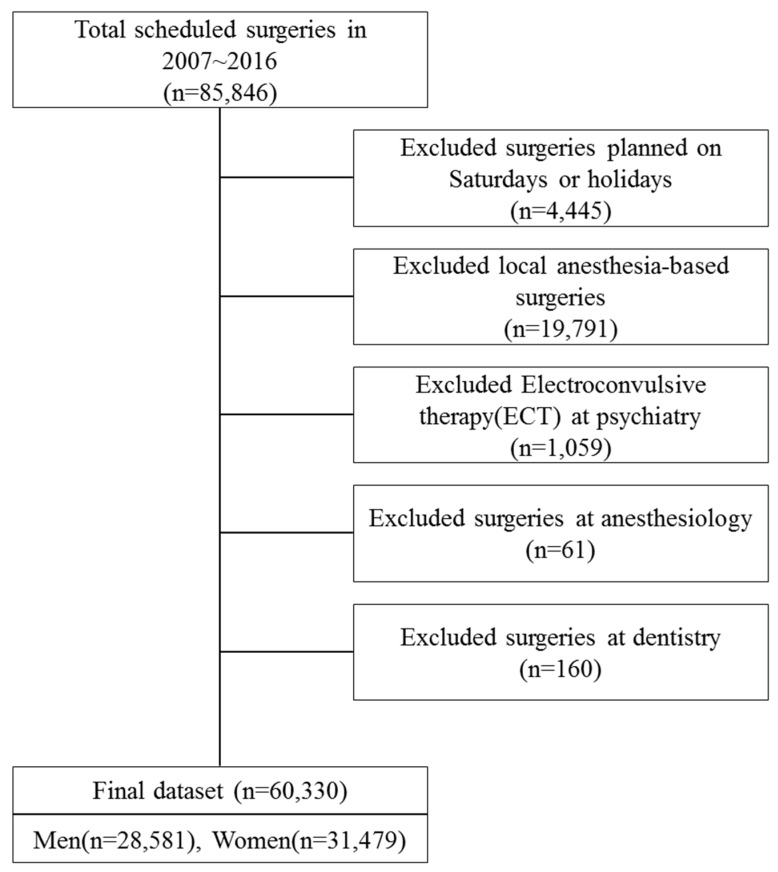
Flowchart for selecting participants.

**Table 1 ijerph-16-00007-t001:** General, Surgery and operation schedule characteristics of study participants.

Variables			Total	Surgery		Cancellation		*p*-Value
				N	%	N	%	
General characteristics	Sex	Male	28,851	26,258	91.0	2593	9.0	<0.0001
		Female	31,479	29,238	92.9	2241	7.1	
	Age	≤19	9183	8429	91.8	754	8.2	<0.0001
		20–39	14,376	13,512	94.0	864	6.0	
		40–59	20,887	9227	92.1	1660	8.0	
		60–79	13,850	2529	90.5	1321	9.5	
		≥80	2034	1799	88.5	235	11.6	
	Admission type	Outpatient/Day surgery clinic	2117	2064	97.5	53	2.5	<0.0001
		Inpatient	58,213	53,432	91.8	4781	8.2	
	Chronic Disease	N/A	44,233	40,867	92.4	3366	7.6	<0.0001
		Hypertension	8775	8040	91.6	735	8.4	
		Diabetes	2543	2297	90.3	246	9.7	
		Both	4779	4292	89.8	487	10.2	
Surgery characteristics	Department	OS (Orthopedic surgery)	18,694	16,883	90.3	1811	9.7	<0.0001
		GS (General surgery)	10,364	9604	92.7	760	7.3	
		OB&GY (Obstetrics & Gynecology)	7728	7427	96.1	301	3.9	
		NS (Neurosurgery)	3023	2706	89.5	317	10.5	
		CS (Thoracic and Cardiovascular Surgery)	2133	1936	90.8	197	9.2	
		PS (Plastic Surgery)	2216	2012	90.8	204	9.2	
		ENT (Ear, Nose & Throat)	8985	8331	92.7	654	7.3	
		EY (Ophthalmology)	3214	3038	94.5	176	5.5	
		UR (Urology)	3973	3559	89.6	414	10.4	
	Type of anesthesia	General anesthesia	43,638	40,387	92.6	3251	7.5	<0.0001
		Regional anesthesia	16,692	15,109	90.5	1583	9.5	
	Emergency	Emergency	10,383	9962	96.0	421	4.1	<0.0001
		Planned	49,947	45,534	91.2	4413	8.8	
	Diagnosis	Injury, poisoning and certain other consequences of external causes (S00-T98)	10,969	9851	89.8	1118	10.2	<0.0001
		Certain infectious and parasitic diseases (A00-B99)	135	114	84.4	21	15.6	
		Neoplasms (C00-D48)	7457	6992	93.8	465	6.2	
		Endocrine, nutritional and metabolic diseases (E00-E90)	443	375	84.7	68	15.4	
		Diseases of the nervous system (G00-G99)	888	785	88.4	103	11.6	
		Diseases of the eye and adnexa (H00-H59)	2702	2587	95.7	115	4.3	
		Diseases of the ear and mastoid process (H60-H95)	1727	1563	90.5	164	9.5	
		Diseases of the circulatory system (I00-I99)	2553	2297	90.0	256	10.0	
		Diseases of the respiratory system (J00-J99)	5266	4929	93.6	337	6.4	
		Diseases of the digestive system (K00-K93)	7381	6841	92.7	540	7.3	
		Diseases of the skin and subcutaneous tissue (L00-L99)	474	422	89.0	52	11.0	
		Diseases of the musculoskeletal system and connective tissue (M00-M99)	8955	8164	91.2	791	8.8	
		Diseases of the genitourinary system (N00-N99)	5470	5049	92.3	421	7.7	
		Pregnancy, childbirth and the puerperium (O00-O99)	2674	2605	97.4	69	2.6	
		Certain conditions originating in the perinatal period (P00-P96)	56	48	85.7	8	14.3	
		Congenital malformations, deformations and chromosomal abnormalities (Q00-Q99)	1367	1219	89.2	148	10.8	
		Symptoms, signs and abnormal clinical and laboratory findings, NEC (R00-R99)	383	294	76.8	89	23.2	
Surgery schedule characteristics	Year	2007	4694	4302	91.7	392	8.4	0.4562
		2008	5320	4891	91.9	429	8.1	
		2009	5227	4819	92.2	408	7.8	
		2010	5695	5210	91.5	485	8.5	
		2011	6307	5826	92.4	481	7.6	
		2012	6483	5968	92.1	515	7.9	
		2013	6383	5873	92.0	510	8.0	
		2014	6605	6040	91.5	565	8.6	
		2015	6573	6071	92.4	502	7.6	
		2016	7043	6496	92.2	547	7.8	
	Season	spring	14,467	13,345	92.2	1122	7.8	0.5001
		Summer	15,219	13,965	91.8	1254	8.2	
		Fall	14,270	13,126	92.0	1144	8.0	
		Winter	16,374	15,060	92.0	1314	8.0	
	Day of week	Mon	12,003	10,909	90.9	1094	9.1	<0.0001
		Tue	12,999	12,039	92.6	960	7.4	
		Wed	12,123	11,164	92.1	959	7.9	
		Thu	11,751	10,833	92.2	918	7.8	
		Fri	11,454	10,551	92.1	903	7.9	
Total			60,330	55,496	92.0	4834	8.0	

**Table 2 ijerph-16-00007-t002:** The reasons for surgery cancellation.

Reason		Total		Planned Surgery		Emergency Surgery	
		N	%	N	%	N	%
Patient initiated cancellation	Additional treatment and screening before surgery	1566	32.4	1414	32	152	36.1
	Patient refused surgery	1444	29.9	1351	30.6	93	22.1
	Delay due to personal reason	950	19.7	921	20.9	29	6.9
	Incomplete Surgical-work up	66	1.4	56	1.3	10	2.4
	Change in medical status	355	7.3	280	6.3	75	17.8
	Change in treatment plan	57	1.2	44	1	13	3.1
	Others	67	1.4	64	1.5	3	0.7
	subtotal	4505	93.2	4130	93.6	375	89.1
Hospital initiated cancellation	Surgeon not available	65	1.3	56	1.3	9	2.1
	Operation in another place	7	0.1	6	0.1	1	0.2
	Schedule error	23	0.5	16	0.4	7	1.7
	Lack of beds	28	0.6	20	0.5	8	1.9
	Equipment not available	10	0.2	7	0.2	3	0.7
	Incomplete Medical evaluation	161	3.3	149	3.4	12	2.9
	Others	35	0.7	29	0.7	6	1.4
	subtotal	329	6.8	283	6.4	46	10.9
Total		4834	100	4413	100	421	100

**Table 3 ijerph-16-00007-t003:** Multivariable regression of the association with surgery cancellation and related factors (All Surgeries).

Variables		OR	95% CI	
Sex	Male	1.00		
	Female	0.87	0.82	0.93
Age	≤19	1.00		
	20–39	0.76	0.68	0.85
	40–59	0.89	0.80	0.98
	60–79	1.04	0.93	1.16
	≥80	1.35	1.14	1.59
Admission type	Outpatient/Day surgery clinic	1.00		
	Inpatient	2.71	2.03	3.61
Chronic Disease	N/A	1.00		
	Hypertension	0.90	0.82	0.99
	Diabetes	1.13	0.98	1.30
	Both	1.10	0.98	1.22
Department	OS (Orthopedic surgery)	1.00		
	GS (General surgery)	0.90	0.82	0.99
	OB & GY (Obstetrics & Gynecology)	0.53	0.46	0.60
	NS (Neurosurgery)	1.39	1.21	1.59
	CS (Thoracic and Cardiovascular Surgery)	0.98	0.84	1.15
	PS (Plastic Surgery)	1.06	0.90	1.24
	ENT (Ear, Nose & Throat)	0.75	0.68	0.84
	EY (Ophthalmology)	0.66	0.56	0.79
	UR (Urology)	1.08	0.97	1.21
Type of anesthesia	General anesthesia	1.00		
	Regional anesthesia	1.15	1.07	1.24
Emergency	Emergency	1.00		
	Planned	2.45	2.21	2.73
Season	spring	1.00		
	Summer	1.08	1.00	1.18
	Fall	1.05	0.96	1.14
	Winter	1.03	0.95	1.12
Day of week	Mon	1.00		
	Tue	0.79	0.72	0.87
	Wed	0.87	0.79	0.96
	Thu	0.86	0.79	0.95
	Fri	0.86	0.78	0.94

**Table 4 ijerph-16-00007-t004:** Multivariable regression of the association with surgery cancellation and related factors (Planned Surgeries).

Variables		OR	95% CI	
Sex	Male	1.00		
	Female	0.88	0.82	0.94
Age	≤19	1.00		
	20–39	0.76	0.68	0.85
	40–59	0.86	0.77	0.96
	60–79	0.98	0.87	1.10
	≥80	1.20	1.00	1.45
Admission type	Outpatient/Day surgery clinic	1.00		
	Inpatient	3.24	2.37	4.44
Chronic Disease	N/A	1.00		
	Hypertension	0.91	0.83	1.00
	Diabetes	1.10	0.95	1.28
	Both	1.11	0.99	1.25
Department	OS (Orthopedic surgery)	1.00		
	GS (General surgery)	1.01	0.92	1.12
	OB & GY (Obstetrics & Gynecology)	0.53	0.46	0.61
	NS (Neurosurgery)	1.63	1.41	1.89
	CS (Thoracic and Cardiovascular Surgery)	0.99	0.84	1.17
	PS (Plastic Surgery)	0.99	0.83	1.17
	ENT (Ear, Nose & Throat)	0.75	0.67	0.84
	EY (Ophthalmology)	0.68	0.57	0.82
	UR (Urology)	1.08	0.96	1.22
Type of anesthesia	General anesthesia	1.00		
	Regional anesthesia	1.14	1.05	1.23
Season	spring	1.00		
	Summer	1.06	0.97	1.16
	Fall	1.03	0.94	1.13
	Winter	1.01	0.92	1.10
Day of week	Mon	1.00		
	Tue	0.80	0.73	0.88
	Wed	0.90	0.82	0.99
	Thu	0.89	0.81	0.98
	Fri	0.88	0.80	0.97

**Table 5 ijerph-16-00007-t005:** Multivariable regression of the association with surgery cancellation and related factors (Emergency Surgeries).

Variables		OR	95% CI	
Sex	Male	1.00		
	Female	0.85	0.68	1.06
Age	≤19	1.00		
	20–39	0.72	0.50	1.05
	40–59	1.10	0.78	1.53
	60–79	1.69	1.17	2.45
	≥80	2.93	1.83	4.69
Admission type	Outpatient/Day surgery clinic	1.00		
	Inpatient	0.42	0.21	0.86
Chronic Disease	N/A	1.00		
	Hypertension	0.82	0.60	1.12
	Diabetes	1.30	0.84	2.02
	Both	0.94	0.65	1.35
Department	OS (Orthopedic surgery)	1.00		
	GS (General surgery)	0.37	0.27	0.51
	OB & GY (Obstetrics & Gynecology)	0.44	0.27	0.70
	NS (Neurosurgery)	0.53	0.37	0.78
	CS (Thoracic and Cardiovascular Surgery)	0.84	0.49	1.43
	PS (Plastic Surgery)	1.40	0.92	2.14
	ENT (Ear, Nose & Throat)	0.63	0.37	1.09
	EY (Ophthalmology)	0.77	0.37	1.63
	UR (Urology)	1.03	0.73	1.45
Type of anesthesia	General anesthesia	1.00		
	Regional anesthesia	1.23	0.95	1.59
Season	spring	1.00		
	Summer	1.30	0.98	1.73
	Fall	1.21	0.90	1.63
	Winter	1.19	0.89	1.61
Day of week	Mon	1.00		
	Tue	0.76	0.57	1.01
	Wed	0.62	0.46	0.84
	Thu	0.66	0.49	0.90
	Fri	0.65	0.48	0.88

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
