# Peer review of "Reasons for Surgery Cancellation in a General Hospital: A 10-year Study"

_ijerph, 2018, doi:10.3390/ijerph16010007_

Round 1
Reviewer 1 Report
This study has the aim to investigate the factors leading to surgery cancellations and the reasons for these cancellations. In my opinion, this is a relevant objective and the methods applied are adequate. However, this study focuses on several variables do not related to the reasons for cancellations. These reasons are briefly described.
The study is about when cancellations are done during the week and a description of the number of cancellations related to gender or age. So, the tittle and the objective could be modified or the text changed.
The information provided about cancellations is adequate to understand underlying factors affecting this issue. However, the study of the causes of these cancellations is appreciate.
The hospital characteristics must be described to understand the data.
Summary needs a deep review and relevant data must be included.
Objective must not include aspects related to methodology.
Surgeries planned on Saturdays and holidays were not included but a justification of this decision is not included.
2.2 Variables. The phrase “The dependent variable in this study was surgery cancellation” does not include information useful. In my opinion could be erased.
Chronic diseases include more than diabetes and hypertension for example COPD or atrial fibrillation or other diseases. Please, introduced a justification of the criteria used.The reasons of cancellations classification are not described in the Methods Section
Dummies variables are not described.
Two decimals are adequate, avoid three or more in table.
Lastly, there is no information about if this study has been approved by the ethical committee of the hospital.
Author Response
1. This study has the aim to investigate the factors leading to surgery cancellations and the reasons for these cancellations. In my opinion, this is a relevant objective and the methods applied are adequate. However, this study focuses on several variables do not related to the reasons for cancellations. These reasons are briefly described.
Answer: We would like to express our gratitude to the reviewer for taking the time to suggest changes to our manuscript. We fully agree with your comment. As you have suggested, the purpose of this study is to analyze the causes of surgery cancellation in relation to various characteristics. Based on this study, hospital staff should be able to reduce the wasting of hospitals’ medical resources by considering factors affecting surgery cancellation. Therefore, we thought it necessary to consider all possible situations rather than focusing on specific variables. We tried to identify factors influencing surgery cancellation by using all the variables available from EMR data. However, other reviewers pointed out variables that did not have a significant effect on the analysis, so we selected only the actual influential variables by the stepwise selection option when performing logistic regression.
2. The study is about when cancellations are done during the week and a description of the number of cancellations related to gender or age. So, the tittle and the objective could be modified or the text changed.
Answer: We appreciate your valuable suggestion. As you mentioned, surgery cancellation was influenced by age and gender. These variables, however, are not the variables of interest, although they did affect surgery cancellation. Therefore, we thought it would be problematic to alter the text or title to focus more on those characteristics. Respecting your opinion, we will carry out a further study on surgery cancellation dependent on patient characteristics.
3. The information provided about cancellations is adequate to understand underlying factors affecting this issue. However, the study of the causes of these cancellations is appreciate.
Answer: Thank you for acknowledging the fact that our study attempts to share with potential readers the causes of cancelations of surgery. We hope the added information helps to enhance the understanding of related study topics.
4. The hospital characteristics must be described to understand the data.
Answer: Thank you for your meaningful suggestion. We totally agree with your comment and have added study hospital characteristics in the data collection paragraph of the Methods section. (Page 2, 40-44 lines)
5. Summary needs a deep review and relevant data must be included.
Answer: We appreciate your valuable comment. We have revised the data and statistical analysis parts of the methods paragraph in the abstract, following your suggestion. (Page 1, 16-22 lines)
6. Objective must not include aspects related to methodology.
Answer: Thank you for your valuable suggestion. The content described in the study objective was a research direction rather than a methodology. However, as you have suggested, we made the objective more concise and relevant. (Page 2, 35-37 lines)
7. Surgeries planned on Saturdays and holidays were not included but a justification of this decision is not included.
Answer: Thank you for your comment. When analyzing the data, the reason for excluding Saturdays and holidays from the surgery schedule is that these are not normal working days. In Korea, general surgery is not planned on Saturdays or holidays unless it is an emergency operation required by the patient's condition. Therefore, we simply stated that we excluded these days in the text and have decided that no further explanation was necessary.
8. 2.2 Variables. The phrase “The dependent variable in this study was surgery cancellation” does not include information useful. In my opinion could be erased.
Answer: Thank you for your meaningful comment. This sentence was originally included to explain the dependent variable in the study. However, we totally agree with your comment and have omitted the sentence.
9. Chronic diseases include more than diabetes and hypertension for example COPD or atrial fibrillation or other diseases. Please, introduced a justification of the criteria used.
Answer: Thank you for your thoughtful comment. The most common chronic diseases among Koreans are diabetes and hypertension. If blood sugar or blood pressure is not controlled, it may affect anesthetic management before the operation or medical outcomes after the surgery, and it may be difficult to perform the operation itself. Therefore, identification and management of diabetes and hypertension in patients scheduled for surgery are very important. A phrase has been added to the text that outlines the inclusion of only these two diseases. (Page 3, 17-20 lines)
10. The reasons of cancellations classification are not described in the Methods Section.
Answer: Thank you for your meaningful comment. We totally agree and have added content to increase understanding in this regard. (Page 4, 17-23 lines)
11. Dummies variables are not described.
Answer: We appreciate your valuable comment. In this study, an explanation of the dummy variables was not considered to be meaningful and, therefore, was not described in the text. In particular, we considered all the variables that affect surgery cancellation, so it was acceptable to explain only the variables used in the analysis.
12. Two decimals are adequate, avoid three or more in table.
Answer: We appreciate your valuable suggestion. In the table, one decimal place was used in figures denoting probability; two decimal places were used for the ORs and 95% CIs in the regression analysis results. The P-values are expressed in 4 decimals to judge the significance of the analysis.
13. Lastly, there is no information about if this study has been approved by the ethical committee of the hospital.
Answer: Thank you for your valuable comment. This study was approved by the institutional review board from Dongguk University Ilsan Hospital (IRB approval; DUIH 2017-06-003-002-HE002) and we added this content in the paragraph, “2.3. Statistical analysis.” (Page 4, 37-38 lines)

Reviewer 2 Report
This is an interesting study with important results, based on a consistent sample of subjects. The authors correctly acknowledge the several study limitations..
I have a major remark which regards the subgroup analysis by type of surgery (section 3.3.2). These groups were already included in the main multivariate analysis and thus this analysis is not useful, also because based on a reduced sample of subjects. Rather, I recommend to run a stepwise analysis (backward or forward) in order to identify the best determinants of surgery cancellation and remove those not important.
Abstract. “As the results of multivariable.... , and it was on monday“. I recommend to enter odds ratio near to each significant determinant of surgery cancellation.
Author Response
This is an interesting study with important results, based on a consistent sample of subjects. The authors correctly acknowledge the several study limitations..
1. I have a major remark which regards the subgroup analysis by type of surgery (section 3.3.2). These groups were already included in the main multivariate analysis and thus this analysis is not useful, also because based on a reduced sample of subjects. Rather, I recommend to run a stepwise analysis (backward or forward) in order to identify the best determinants of surgery cancellation and remove those not important.
Answer: We appreciate the constructive comments regarding our study. In this study, we investigated the factors influencing surgery cancellation by using all the possible variables from EMR data. Therefore, in the logistic regression, we have already identified important determinants through the stepwise selection function in the model option, as you suggested, and presented only the significant results. The base model included the month of the scheduled surgery and a detailed type of anesthesia (general anesthesia, spinal anesthesia, epidural anesthesia, MAC, local anesthesia, nerve block, other), in addition to other variables in the study results. We appreciate your suggestion and have added a relevant section to the paragraph, “2.3. Statistical analysis.” (Page 4, 33-34 lines)
2. Abstract. “As the results of multivariable.... , and it was on monday“. I recommend to enter odds ratio near to each significant determinant of surgery cancellation.
Answer: Thank you for your comment and we completely agree with you. According to your suggestion, we added the odds ratios for the logistic regression results and revised the content in the results portion of the abstract. (Page 1, 22-29 lines)

Round 2
Reviewer 1 Report
All suggestions have been adequately replayed
Reviewer 2 Report
I have seen the revised version. Although I am still skeptic about the appropriateness of the subgroup analysis with logistic regression (section 3.3.2), the authors have adequately answered my remarks.